# Unsupervised SE(3) Disentanglement for *in situ* Macromolecular Morphology Identification from Cryo-Electron Tomography

## Abstract

Cryo-electron tomography (cryo-ET) provides direct 3D visualization of macromolecules inside the cell, enabling analysis of their *in situ* morphology. This morphology can be regarded as an $SE(3)$-invariant, denoised volumetric representation of subvolumes extracted from tomograms. Inferring morphology is therefore an inverse problem of estimating both a template morphology and its SE(3) transformation. Existing maximum likelihood–based solution to this problem often miss rare but important morphologies and require extensive manual hyperparameter tuning. Addressing this issue, we present a disentangled deep representation learning framework that separates SE(3) transformations from morphological content in the representation space. The framework includes a novel multi-choice learning module that enables this disentanglement for highly noisy cryo-ET data, and the learned morphological content is used to generate template morphologies. Experiments on simulated and real cryo-ET datasets demonstrate clear improvements over prior methods, including the discovery of previously unidentified macromolecular morphologies.

## 1 Introduction

Over the last decade, structural biology has shifted towards morphological characterization of large macromolecular complexes and assemblies, particularly in a near-native *in situ* environment (Turk & Baumeister, 2020). Cellular cryo-electron tomography (cryo-ET) has played a pivotal role in enabling the paradigm shift, serving as a practical tool for 3D visualization of macromolecule shape and morphology in their native states within the cell (Doerr, 2017; Turk & Baumeister, 2020). In this imaging technique, a specimen of a whole cell or part of a cell is placed under an electron microscope and images are captured across different tilt angles (typically from $-60°$ to $+60°$). To prevent radiation damage, the electron dosage is kept at a low level. The low electron dosage, along with the crowded cytoplasmic environment, makes the cryo-ET images extremely noisy. The tilt-series cryo-ET images are reconstructed into large 3D grayscale volumes, known as 3D tomograms. The tomograms are very large volumetric arrays (typically in the range of $4000 \times 4000 \times 1000$ voxels), typically containing hundreds to thousands of structurally heterogeneous macromolecular complexes in diverse orientations, along with other subcellular objects, including organelles and membranes.

The cryo-ET macromolecular structure processing workflow first extracts small subvolumes from a tomogram that potentially contains a macromolecule, known as subtomograms (Chen et al., 2019; Scheres, 2012). The extracted subtomograms are further analyzed to identify the morphologies of macromolecular complexes. However, identification of macromolecular morphologies from these subtomograms is a complex process due to numerous challenges, including high noise, structural and orientational heterogeneity, and other imaging artifacts (Turk & Baumeister, 2020) present in the tomograms. Identifying macromolecular morphologies from subtomograms can be formulated as an inverse problem: determining template volumes under the assumption that each subtomogram is generated by applying an $SE(3)$ (or equivalently, $SO(3) \ltimes \mathbb{R}^3$) transformation with unknown parameters to an unknown template volume. The most traditional approach to solve this inverse problem involves performing maximum-likelihood based subtomogram classification and initial template generation (Chen et al., 2019). In this approach, each subtomogram is assigned to a class and trans-

formation probabilistically based on the estimated template volume of the class, which is updated iteratively along with the transformations (Scheres, 2012). Thus, a template volume is estimated for each subtomogram, which denotes its coarse morphology. To obtain a fine-grained morphology, the coarse templates are refined to an optimal resolution in the follow-up subtomogram averaging (STA) or sub-tilt reconstruction step (Chen et al., 2019). Nevertheless, the overall morphology is identified in the subtomogram classification and initial template generation step. This approach, despite being the go-to method for decades, is often unable to resolve rare but crucial morphologies. Moreover, performance highly depends on manually setting appropriate values for a large number of hyperparameters.

In this work, we addressed this decade-long never-before-solved problem with a novel unsupervised deep learning-based method. Our approach is automated and does not require users to adjust a large number of hyperparameters, unlike the maximum likelihood-based method. Using a disentangled representation learning (DRL) framework called Harmony (Uddin et al., 2022), it maps each subtomogram to a disentangled $SE(3)$ transformation space and morphology latent space (Figure 1). A generator network conditioned only on the morphology latent space is used to identify the template volume or macromolecular morphology in a subtomogram. However, for very low-SNR realistic subtomograms, simply tailoring $SE(3)$ disentanglement does not result in satisfactory morphology identification performance (Table 1). To solve this issue, we introduced a novel multi-choice learning based approach. In this approach, the DRL framework is presented with multiple choices of differently transformed template volumes. The framework then uses the most optimal choice to minimize its objective function via a 'winner-takes-all' loss (Figure 1). With this multi-choice loss coupled with $SE(3)$ disentanglement, our method effectively identifies morphologies given realistic subtomograms.

We tested our method against several simulated datasets with different levels of signal-to-noise (SNR) ratios and imaging artifacts. Our method showed significantly superior performance compared to the existing maximum likelihood-based approach. Our experiments also revealed the importance of our novel multi-choice learning module on top of $SE(3)$ disentanglement. We finally validated our method against subtomograms of the thylakoid membrane region in publicly available *Chlamy* cellular cryo-ET images, where our method identified several morphologies previously unidentified with existing approaches. We anticipate that our method can serve as a useful tool and an alternative or complement to existing maximum-likelihood based approaches to determine morphologies of numerous macromolecular complexes of previously unknown structures inside their native cellular context.

We summarize our contributions as follows:

- We developed a novel unsupervised learning-based method to solve a crucial and largely unsolved problem of *in situ* morphology identification of macromolecules inside the cell from cellular cryo-ET images.

- We developed a novel multichoice learning module that enables unsupervised $SE(3)$ disentanglement for real cryo-ET datasets that typically have an extremely low signal-to-noise ratio.

- We identified several macromolecular morphologies in real cell cryo-ET subtomograms of thylakoid membranes that were previously undisclosed by existing approaches.

## 2 RELATED WORKS

**Macromolecule identification in cellular Cryo-ET:** Identifying macromolecules inside cellular cryo-electron tomograms has been an open and crucial challenge for several decades (Turk & Baumeister, 2020; Uddin et al., 2025b). The earliest and most traditional approach is template matching (Böhm et al., 2000) with known structural templates to identify these macromolecules within the cell. However, this approach is prone to template-specific biases and cannot identify novel morphologies that lack a known template (Zeng et al., 2023). Subsequently, supervised segmentation-based approaches (Moebel et al., 2021; Liu et al., 2024) gained popularity to identify macromolecules in cellular cryo-ET data. These methods are particularly effective when many copies of a given macromolecule have already been manually annotated, since the availability of such annotations provides the training datasets required for reliable model supervision. However, the

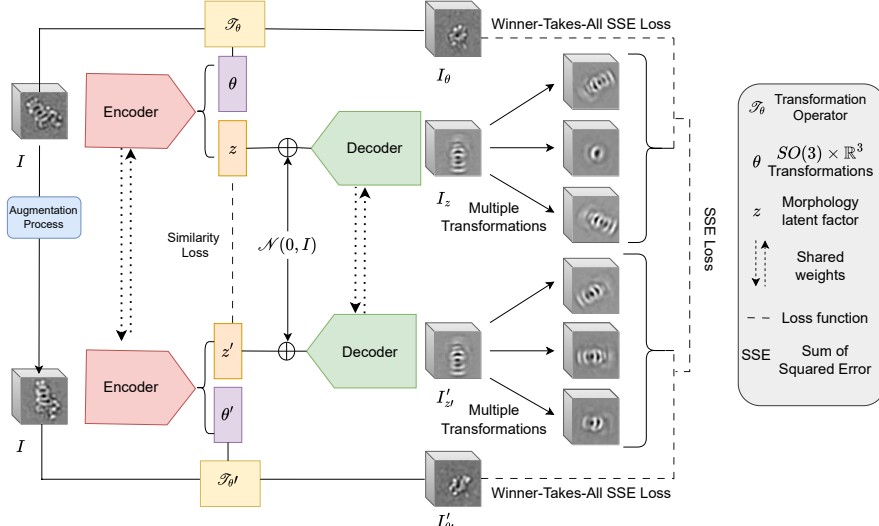

Figure 1: **Schematic overview of our method**. Input subtomograms and their augmentations are encoded into disentangled latent factors: transformation parameters ($\theta$) and morphology factors ($z$). Decoders reconstruct the input under multiple transformations, with a winner-takes-all Sum of Squared Error (SSE) loss selecting the best reconstruction. A similarity loss enforces consistency between augmented pairs, encouraging separation of transformation and morphology.

requirement of manual annotation itself poses a great challenge. It also depends on prior knowledge of structures and is limited in identifying macromolecules with unknown morphologies. Furthermore, because of the crowded and noisy nature and large size of the cellular tomograms, annotations are extremely burdensome and often impossible for macromolecules of small size.

Consequently, template-free unsupervised methods for identifying macromolecules have been adopted. We provide further details on this setup in the Appendix A.2. Here, small subvolumes called 'subtomograms' containing peak signals are extracted from cellular tomograms, and macromolecules are identified from these subvolumes in an inverse problem manner. Hence, the subvolumes are assumed to have resulted from a forward process of selecting a template from a number of templates, rotating and translating the template in 3D, and then applying image optics and noise effects. The structural templates and the corresponding transformations remain unknown and are estimated from the subvolumes. The most common approach to this problem is a maximum-likelihood-based 3D classification and template generation approach in RELION (Scheres, 2012), which has been discussed in detail in the Introduction. Similar to RELION, our method also identifies macromolecules from subvolumes in a template-free, unsupervised manner. However, instead of a maximum-likelihood-based approach, our method uses deep representation learning with SE(3) disentanglement. In addition, our method does not require users to manually define optimal values for a large number of hyperparameters similar to RELION. A learning-based unsupervised solution called DISCA (Zeng et al., 2023) has been developed recently that performs subtomogram classification followed by RELION classification and averaging to identify macromolecular morphologies inside the cell. However, a large dependence on RELION remains. Unlike DISCA, our method does not depend on RELION classification to generate templates. Furthermore, DISCA aims to learn SE(3) invariant features for transformation using a sophisticated network architecture. Unlike it, we perform SE(3) disentanglement with much simple encoder-decoder framework.

**Multi-choice learning (MCL):** Multi-choice learning (MCL) (Guzman-Rivera et al., 2012) refers to generating multiple choices or hypotheses for the model and making it learn from the most optimal choice. In this learning paradigm, a 'winner-takes-all' loss is used, where the loss is optimized through the most accurate model choice. This is different from standard mixture-of-expert setting where a weighted combination of the model choices are optimized. MCL has been used to reduce ambiguity in several machine learning prediction tasks, including image segmentation (Kohl et al., 2018), human pose and shape estimation (Biggs et al., 2020), motion forecasting (Yuan & Kitani), etc. Very recently, CryoSPIN (Shekarforoush et al., 2024) has used MCL to estimate pose from 2D

cryo-EM single-particle images for the cryo-EM *ab initio* reconstruction task (Levy et al., 2025; Rangan et al., 2024). The cryo-EM images are 2D projections of an underlying 3D volume, which is estimated in the reconstruction task. Unlike this work, we disentangle SE(3) transformation from 3D cryo-ET subvolumes, which are much noisier than their 2D cryo-EM counterparts. Moreover, (Shekarforoush et al., 2024) uses a multi-head encoder that generates 4 choices of $SO(3)$ rotations, and the model selects the most optimal one. Unlike this approach, we generate multiple choices of template volumes with varying $SE(3)$ transformations of the generator output, and let the model select the most optimal template volume.

**Disentangled transformation representation learning:** Disentangled representation of transformations and content is a fundamental part of our method. SpatialVAE (Bepler et al., 2019), Harmony(Uddin et al., 2022), and VITAE (Skafte & Hauberg, 2019) are a few methods that perform explicit disentanglement of transformations from semantic content in visual data. In such methods, the transformation and content factors are explicitly separated in the latent representation space. Among them, SpatialVAE performs disentanglement of SE(2) transformations by explicitly parameterizing the SE(2) transformation in the latent space and conditioning image generation on the transformed coordinate frame in pixel-by-pixel format. VITAE performs disentanglement of 2D diffeomorphic transformations from semantic content by a specialized parameterization of transformation latent space. Harmony uses self-supervised learning to disentangle transformations with any parameteric functional form, including but not limited to SE(2) and SE(3) transformations. In this work, we build on the Harmony framework to specifically disentangle SE(3) transformations. Unlike Harmony, which serves as a general-purpose disentanglement framework, our approach is tailored to macromolecular morphology identification in cellular cryo-ET data, incorporating novel loss functions and task-specific mechanisms.

**Single-particle cryo-EM reconstruction methods:** Several deep learning based generative models, such as, cryoDRN (Zhong et al., 2021), e2GMM (Chen & Ludtke, 2021), cryoSPARC-3DVA (Punjani et al., 2017), cryoAI (Levy et al., 2022), cryoSPIN (Shekarforoush et al., 2024), etc., have been developed in recent years that identies multiple conformations of macromolecules from 2D single-particle cryo-EM images. A few of them (cryoAI, cryoSPIN) does not require predefined poses and instead optimizes them directly. CryoDRGN and cryoAI has been further extended for subtilt-image reconstruction in cryo-ET, named CryoDRGN-ET (Rangan et al., 2024) and CryoDRGN-AI-ET (Levy et al., 2025). However, these methods still applies to single particle cryo-ET tomograms primarily containing macromolecules of nearly homogeneous morphologies. They are not applicable for identifying highly heterogeneous morphologies from subtomogram mixtures. In Appendix A.1, we discussed about this issue in details. Unlike these 2D projection dependent methods, our method identifies highly heterogeneous 3D morphologies directly from 3D subtomogram data.

## 3 METHODS

### 3.1 PROBLEM DEFINITION

Consider a set of cryo-ET subtomograms $\{I_i\}_{i=1}^N$, $I_i \in \mathbb{R}^{d \times d \times d}$, where $d \in \mathbb{N}$ is the dimension of the subtomogram. Each subtomogram $I_i$ is assumed to be generated from an unknown template volume $V_i \in \{V_k\}_{k=1}^K, V_i \in \mathbb{R}^{d \times d \times d}$, where $K \in \mathbb{N}$ is fixed. The generative process is modeled as the action of a rigid-body transformation in the special Euclidean group $SE(3)$ or $SO(3) \ltimes \mathbb{R}^3$, followed by convolution with a point spread function and additive noise. Formally,

$$I_i = g(S_{t_i} R_{\phi_i} V_i) + \eta_i, \qquad i = 1, \dots, N,$$

where

- $R_{\psi_i} \in SO(3)$ denotes a rotation operator parameterized by $\phi_i$,
- $S_{t_i}$ denotes a translation operator parameterized by $d_i \in \mathbb{R}^3$,
- $g$ is the cryo-ET imaging operator (e.g., convolution with the point spread function).
- $\eta_i$ is a noise term modeling imaging artifacts.

The problem is modeled as an inverse problem, where given only the observed subtomograms $\{I_i\}_{i=1}^N$ and the number of templates $K$, the task is to estimate the set of template volumes $\{V_k\}_{k=1}^K$

together with the latent transformation parameters $\{\theta_i\}_{i=1}^N = \{(\phi_i, t_i)\}_{i=1}^N \subset SO(3) \ltimes \mathbb{R}^3 \cong SE(3)$.

### 3.2 DISENTANGLING $SE(3)$ FROM MACROMOLECULAR MORPHOLOGY IN SUBTOMOGRAMS

#### 3.2.1 UNSUPERVISED $SE(3)$ DISENTANGLEMENT

Our method performs unsupervised $SE(3)$ disentanglement to identify macromolecular morphologies from cryo-ET subtomograms. To this end, it uses an autoencoder-like DRL framework, Harmony (Uddin et al., 2022), inspired by Siamese training strategy. In this framework, an input image $I$ and its augmented counterpart $I'$ are passed through a shared encoder (Figure 1). The encoder outputs a semantic latent factor $z$ and a $SE(3) \cong SO(3) \ltimes \mathbb{R}^3$ transformation parameter vector $\theta$ for the input image $I$. Similarly, the encoder outputs $z'$ and $\theta'$ for $I'$. The semantic latent vectors $z$ and $z'$ are then passed through a shared decoder to generate two images $I_z$ and $I'_{z'}$, respectively, which reflect transformation-invariant representations of the original input. At the same time, the transformation parameters $\theta$ and $\theta'$ are used to transform $I$ and $I'$, $I_\theta$ and $I'_{\theta'}$, respectively.

To parameterize the $SO(3)$ rotation, we have used the $S2S2$ representation recommended by (Zhou et al., 2019). This parameterization represents the $SO(3)$ rotation with 6 parameters. The 3D $\mathbb{R}^3$ translation is simply represented by 3 parameters, which represent the translation in the $x$, $y$, and $z$ directions. So, the transformation parameter $\theta$ is represented by in total 9 parameters. To create $I'$ from $I$, we use a subtomogram-specific augmentation process instead of applying a random $SE(3)$ transformation. Since subtomograms in a dataset shares the same missing wedge (details in the Appendix), applying a random $SO(3)$ would introduce artificial variations in wedge orientation that do not exist in the data, resulting in unrealistic augmentations. Consequently, we avoid $SO(3)$ and instead apply a random in-plane rotation $SO(2)$ restricted to the $xy$-plane to generate $I'$ from $I$ as a realistic augmentation.

The loss function optimized by the framework consists of two components: (1) a reconstruction loss $L_{\text{recon}}$ that minimizes the sum of squared errors (SSE) between the decoder outputs ($I_z, I'_{z'}$) and the input images transformed by the predicted transformation parameters ($I_\theta, I'_{\theta'}$), and (2) a latent space regularization loss $L_{\text{embed}}$ that penalizes the distance between the semantic latent vectors $z$ and $z'$. The full loss function can be expressed as:

$$L = L_{\text{recon}} + L_{\text{embed}}$$
$$L_{\text{recon}} = \text{SSE}(I_\theta, I_z) + \text{SSE}(I'_{\theta'}, I'_{z'}) + \text{SSE}(I_\theta, I'_{\theta'})$$
$$L_{\text{embed}} = \text{Dis}(z, z')$$

SSE is the sum of the squared error between two images. Dis is a distance metric, which we implemented as the L1 distance between $z$ and $z'$. The entire encoder-decoder architecture is trained end-to-end minimizing this loss function via gradient descent. To enforce a smooth manifold for morphology latent space, additive gaussian noise $\epsilon \sim \mathcal{N}(0, I)$ and $\epsilon' \sim \mathcal{N}(0, I)$ are added to $z$ and $z'$, respectively during training.

#### 3.2.2 MULTI-CHOICE LEARNING FOR $SE(3)$ DISENTANGLEMENT IN SUBTOMOGRAMS

We observed that simply minimizing the above loss does not result in convincing $SE(3)$ disentanglement and morphology modeling for subtomograms, particularly in realistic subtomograms with extremely low SNRs (Table 1). We hypothesize that this issue arises due to the uncertainty in the template generated by the decoder and its relative $SE(3)$ transformation under extremely low signal in real subtomograms. Given the efficacy of multi-choice learning (MCL) in cases of uncertainity in predictions, we incorporated MCL into our $SE(3)$ disentanglement framework.

Instead of directly minimizing the SSE distance between the decoder output and the transformed input as in the original Harmony framework, we present the network with multiple candidates by transforming the decoder output. We then find the candidate best fitting with the transformed input and minimize the SSE distance between them. We refer to this as 'winner-takes-all' SSE loss since the SSE loss is only minimized for the 'winner candidate' having the least distance with the transformed input.

Specifically, we generate $N$ randomly transformed instances $(I_{z_1}, I_{z_2}, \ldots, I_{z_N})$ of each decoder output $I_z$. We apply transformations consisting of uniformly sampled $SO(3)$ rotations and translations within empirically chosen bounds. Instead of simply using $\mathrm{SSE}(I_\theta, I_z)$, we use $\min_{k \in \{1, \ldots, N\}} \mathrm{SSE}(I_\theta, I_{z_k})$ to calculate $L_{\mathrm{recon}}$. We do similar for the decoded output $I'_{z'}$ for the other branch of our siamese network. Overall, our $L_{\mathrm{recon}}$ becomes:

$$L_{\mathrm{recon}} = \min_{k \in \{1, \ldots, N\}} \mathrm{SSE}(I_\theta, I_{z_k}) + \min_{k' \in \{1, \ldots, N\}} \mathrm{SSE}(I'_{\theta'}, I'_{z'_{k'}}) + \mathrm{SSE}(I_\theta, I'_{\theta'})$$

For the first few ($\approx 40$) epochs, we sampled the entire $SO(3)$ grid. In later epochs, we sampled close to the identity matrix in the $SO(3)$ grid. We follow this strategy since after a few epochs of training with sampling of the entire $SO(3)$ grid, the model somewhat learns to decode the optimal structure. We then restrict the $SO(3)$ grid sampling close to the identity matrix. The $SO(3)$ grid sampling is discussed in detail in the Appendix. For translations, we applied shifts up to 2 voxels along each of the $x$, $y$, and $z$ axes.

### 3.3 Inferring morphology of macromolecular complexes from cellular subtomograms

After training the model, we use its encoder and decoder for morphological identification. For classifying the subtomograms in different morphology classes, we first infer the morphological latent factor $z$ for each subtomogram $I$ using the trained encoder network. We then use UMAP (Uniform Manifold Approximation and Projection) to reduce the dimension of the morphological latent factor to 2. We then apply GMM (Gaussian Mixture Model) to cluster the dimension-reduced latent factors into $K$ classes, where $K$ is predefined. Using the GMM model, for each subtomogram, we obtain the probability of it being assigned to any of the $K$ classes. We assign each subtomogram to the class for which it has the highest probability. Thus, we obtain the morphological classification of the subtomograms. Then we use the trained decoder network to visualize the template morphology $V$ of these morphology classes. To obtain representative template for each morphology class, we choose the median of the dimension-reduced semantic latent factor for all subtomograms belonging to that particular class and pass it through the decoder.

## 4 Experiments

**Simulated datasets:** For benchmarking, we created several simulated macromolecule mixture subtomogram datasets. To this end, we collected 4 PDB structures of 4 different macromolecules expressed in yeast cells. These include the 80S ribosome (PDB ID: 4V7R), the 26S proteasome (PDB ID: 3JCP), fatty acid synthase (PDB ID: 2UV8), and TRiC (PDB ID: 7YLU). We filter the PDB structures to 15 Åwith a pixel size of 7.5 Å. We created 1,000 subtomograms from each of the filtered PDB structures. To create the subtomograms, we first performed random $SO(3)$ rotation and small shifts from the center to the filtered PDB structures. Then we apply CTF and noise to match the desired SNR value of the subtomograms. We created subtomograms with SNR 0.1 and SNR 0.01. For each of the SNR levels, we created two sets of subtomograms: one with the missing wedge effect and another without. To create the missing wedge effect, we used a missing-wedge angle (MWA) of $30°$, which is commonly found in experimental subtomograms. Thus, we obtain four simulated subtomogram datasets: 1) SNR 0.1 and missing wedge angle 0, 2) SNR 0.1 and missing wedge angle 30, 3) SNR 0.01 and missing wedge angle 0, and 4) SNR 0.01 and missing wedge angle 30. Dataset 1 has idealistic conditions that are not found in real-world experiments. On the other hand, dataset 4 is the most complex and highly mimics real-world conditions.

**Experimental dataset:** As experimental dataset, we used cell-ET tomograms of *chloromydomonas reinhardtii* algae of EMPIAR-11830. In particular, we studied the structurally heterogeneous and biologically significant region of the thylakoid membrane (Figure 3 A). The thylakoid membrane region contains membrane proteins and several membrane-bound complexes, all of which are morphologically distinct and can be identified in the resolution range of a cryo-ET tomogram (Figure 3B). Given the relatively high signal in the membrane regions, automated picking (Tang et al., 2007) worked reasonably well. We used automated particle picking (Tang et al., 2007) to extract 55,118 subtomograms in the thylakoid membrane regions of *Chlamydomonas reinhardtii*. Unlike simulated datasets, the experimental data set does not contain 'ground truth morphology classes or templates. Consequently, the evaluation on this dataset could be qualitative only.

Table 1: Quantitative comparison of our method and the baselines against the simulated macro-molecule mixture datasets. (↑) indicates the higher score is better. For each experimental setup, we performed three experiments with three different random seeds. We report the mean values in the table. We observed that ARI and Acc varies within $\pm 0.05$ range of mean, SAP varies within $\pm 0.02$ range of mean, and the AUC-FSC scores varies within $\pm 0.005$ range of their mean values.

| Dataset | Method | ARI (↑) | Acc (%) (↑) | SAP (↑) | AUC-FSC (↑) | | | |
|---|---|---|---|---|---|---|---|---|
| | | | | | FAS | Proteasome | Ribosome | TriC |
| SNR 0.1 MWA 0 | RELION | 0.959 | 98.4 | - | 0.537 | 0.514 | 0.544 | 0.535 |
| | CryoDRGN-AI-ET | - | - | - | 0.093 | 0.084 | 0.109 | 0.104 |
| | DISCA + RELION refine | 0.96 | 97 | - | 0.537 | 0.517 | 0.548 | 0.535 |
| | Harmony3D | 0.986 | 99.5 | 0.57 | 0.278 | 0.272 | 0.303 | 0.229 |
| | Our Method | **0.998** | **100** | **0.63** | 0.295 | 0.287 | 0.238 | 0.354 |
| | Our method + RELION refine | **0.998** | **100** | **0.63** | **0.582** | **0.547** | **0.583** | **0.560** |
| SNR 0.1 MWA 30 | RELION | 0.958 | 98.3 | - | 0.508 | 0.467 | 0.494 | 0.502 |
| | CryoDRGN-AI-ET | - | - | - | 0.104 | 0.068 | 0.192 | 0.184 |
| | DISCA + RELION refine | 0.942 | 96.6 | - | 0.510 | 0.470 | 0.500 | 0.521 |
| | Harmony3D | 0.981 | 99.5 | 0.45 | 0.185 | 0.216 | 0.236 | |
| | Our method | **0.989** | **99.6** | **0.60** | 0.286 | 0.254 | 0.126 | 0.281 |
| | Our method + RELION refine | **0.854** | **94.4** | **0.60** | **0.527** | **0.489** | **0.523** | **0.518** |
| SNR 0.01 MWA 0 | RELION | 0.652 | 74.8 | - | 0.520 | 0.147 | 0.160 | 0.498 |
| | CryoDRGN-AI-ET | - | - | - | 0.091 | 0.045 | 0.102 | 0.110 |
| | DISCA + RELION refine | 0.55 | 64 | - | 0.405 | 0.392 | 0.174 | 0.425 |
| | Harmony3D | 0.716 | 89.1 | 0.35 | 0.261 | 0.187 | 0.25 | 0.267 |
| | Our method | **0.913** | **96.7** | **0.49** | 0.242 | 0.232 | 0.278 | 0.229 |
| | Our method + RELION refine | **0.854** | **94.4** | **0.49** | **0.524** | **0.447** | **0.460** | **0.509** |
| SNR 0.01 MWA 30 | RELION | 0.343 | 59.3 | - | 0.469 | 0.111 | 0.120 | 0.140 |
| | CryoDRGN-AI-ET | - | - | - | 0.083 | 0.038 | 0.099 | 0.094 |
| | DISCA + RELION refine | 0.49 | 62 | - | 0.415 | 0.430 | 0.151 | 0.400 |
| (realistic) | Harmony3D | 0.694 | 88.1 | 0.30 | 0.222 | 0.137 | 0.193 | 0.238 |
| | Our method | **0.854** | **94.4** | **0.46** | 0.241 | 0.218 | 0.190 | 0.233 |
| | Our method + RELION refine | **0.854** | **94.4** | **0.46** | **0.472** | **0.447** | **0.483** | **0.470** |

**Training details:** We used a convolutional neural network without pooling layers to implement the encoder and decoder of our model. Detailed discussion of the encoder and decoder network implementation is provided in the Appendix. We trained our model with the Adam optimizer with a constant learning rate of 0.0001 for 200 epochs. We used NVIDIA A5000 GPUs for training our model against the datasets. For the simulated datasets, we trained using a single GPU. For the thylakoid membrane dataset, we trained our model in a distributed manner using two GPUs. We used the PyTorch Accelerate package for the distributed training. Before training, the 3D subtomograms are usually preprocessed (preprocessing details in the Appendix). To implement the MCL module, we use $N = 96$ in our experiments. Overall, $N \geq 64$ gives reasonable performance in the SNR 0.01 setting. For higher SNR idealistic datasets, much lower $N$ ($\leq 10$) is sufficient.

**Evaluation Metrics:** Given that we have ground truth in the simulated dataset, we can perform a quantitative evaluation of our method and RELION (Scheres, 2012) on the simulated dataset. To evaluate clustering performance, we use the adjusted rand index (ARI) and accuracy. For calculating accuracy, we first aligned the predicted unsupervised labels using our approach with the ground-truth labels via Hungarian matching, and then calculated the accuracy between the aligned labels and the ground-truth labels. To assess the quality of the decoder output, we calculated the Area under the Fourier Shell Correlation (FSC) curve (Jeon et al., 2024) with respect to the corresponding templates we used for data simulation. To calculate the SE(3) disentanglement of the latent space, we use SAP score that measures the difference in predictivity of the ground truth morphology classes given the two disentangled latent spaces. Further details on these metrics and their calculation are provided in the Appendix.

## 5 RESULTS

### 5.1 RESULTS IN SIMULATED CELLULAR SUBTOMOGRAM DATA

We begin our experiments with the simulated datasets. We use RELION 3D classification, DISCA, cryoDRGN-AI-ET (Levy et al., 2025), our method without MCL (which we refer to as Harmony3D),

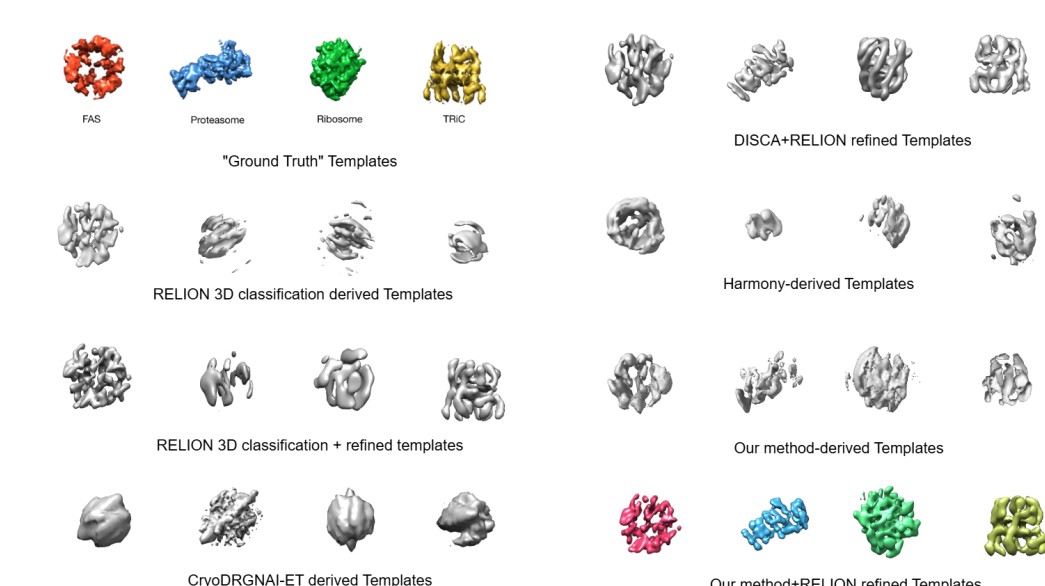

Figure 2: **Results on realistic simulated data (SNR** $0.01$ **and** $30°$ **Missing Wedge Angle)**.

and our complete method (with our MCL module). For all the relevant methods, we set the number of morphology classes, $K = 4$. Among these methods, only cryoDRGN-AI-ET operates on subtilt images, not 3D subtomograms. Hence, we applied cryoDRGN-AI-ET on the subtilt images corresponding to our subtomograms. We used 61 subtilt images with $2°$ interval for each subtomogram. For the other methods, we directly applied on the subtomograms. We quantitatively assessed the classification performance and the template generation performance (Table 1). We further show the templates obtained by refining them with RELION refinement. We investigated the distinct morphology templates obtained by each method in the datasets. We show the templates obtained for the realistic simulated dataset with SNR $0.01$ and MWA $30°$ in Figure 2. We provide the templates obtained for other datasets in the Appendix.

Table 1 shows that our method consistently shows the best classification performance (ARI and accuracy) in all simulated datasets. However, in idealistic data sets with high SNR ($0.1$), the improvement of our method over the baselines is not as significant compared to realistic data sets with low SNR ($0.01$). This suggests the necessity of our method, particularly for real subtomogram datasets, where the performance of other methods is not satisfactory. Furthermore, the large improvement over Harmony3D (our method without MCL) on realistic subtomogram datasets with low SNR suggests the particular efficacy of our proposed MCL module for realistic subtomograms. In fact, for idealistic high SNR dataset, the Harmony3D itself is sufficient. The uncertainty in prediction tasks increases significantly under low SNR of realistic subtomograms, where MCL becomes effective. The AUC-FSC scores also suggest a similar trend.

Figure 2 shows that RELION 3D classification could only somewhat recover the 'ground truth' FAS template on the realistic simulated dataset. In fact, RELION is highly effective in recovering FAS morphology in all the simulatated datasets (Table 1), largely due to the strong symmetric signal present in FAS subtomograms. However, RELION failed to correctly identify other macromolecular complexes, including the ribosome, proteasome, and TRiC, particularly in realistic simulated data, even after performing downstream template refinement. Harmony3D performed even worse, as the generated templates barely resembled the 'ground truth' morphologies (Figure 2). This further underscores the necessity of the proposed MCL module for realistic subtomogram data analysis. CryoDRGNAI-ET (Levy et al., 2025), the method for heterogeneous reconstruction from sub-tilt images, could not identify any of the morphologies, as expected. This further strengthens the claim that sub-tilt reconstruction methods are not suitable for resolving higher degrees of morphological heterogeneity. While the subtomogram classification method, DISCA (Zeng et al., 2023) was able to recover several macromolecular morphologies after RELION refinement, its performance was limited, and considerable improvements were necessary. Finally, with our complete method

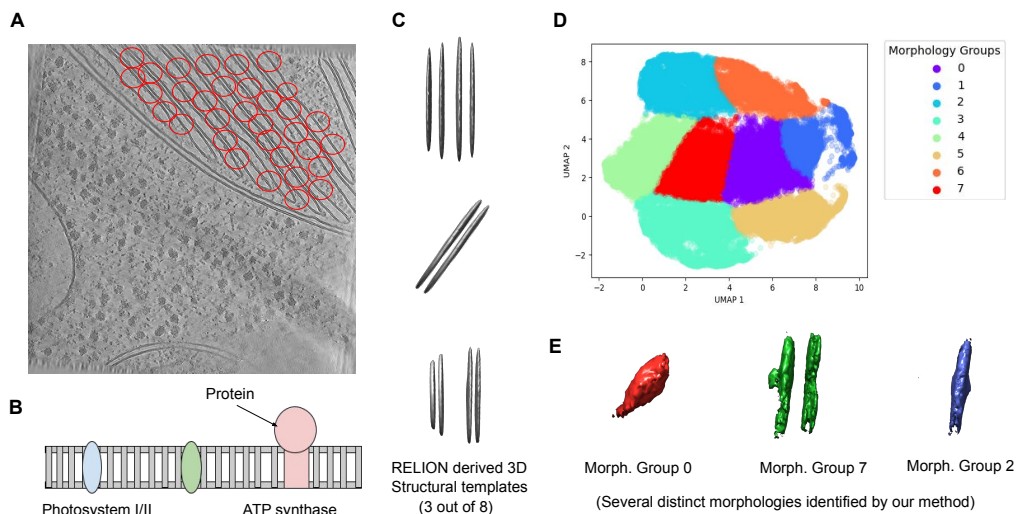

Figure 3: Our method recognizes the morphology of membrane proteins and several membrane-bound enzyme complexes in thylakoid membrane region of *Chlamydomonas reinhardtii*. **A.** A central slab (slice across depth axis) of *Chlamydomonas reinhardtii* tomogram, where subtomograms are extracted from thylakoid membrane region. **B.** Schematic representation of membrane-bound protein complexes involved in photosynthesis in the thylakoid membrane. **C.** A few of the morphologies (initial 3D models) generated by RELION and DISCA. **D.** The UMAP visualization of the morphology latent factor by our method along with the morphology groups obtained with GMM (K=8). **E.** A few of the morphologies (decoded outputs of median of certain morphology classes) obtained by our method.

(Harmony3D + MCL), we achieved markedly improved results, successfully recovering the coarse morphology of all 'ground truth templates. We also obtained fine-grained morphologies by applying downstream refinement to our method outputs.

## 5.2 RESULTS IN EXPERIMENTAL CELLULAR SUBTOMOGRAMS DATA

We first used the RELION 3D classification and DISCA to identify the morphologies present in the experimental dataset. For the 3D classification, we used $K = 8$ expecting that the dataset should not have more than 8 morphologically distinct classes. The RELION 3D classification provided 8 3D structure models. We investigated the structure models with isosurface visualization (Figure 3 C). However, we did not observe any structure densities other than membranes. Similar phenomenon was observed with DISCA classification followed by RELION refinement. Then we applied our method against the subtomogram dataset. After training our unsupervised method on the dataset, we inferred the semantic factors for each subtomogram and calculated the UMAP. We clustered the UMAP with Gaussian Mixture Model (GMM) with $K = 8$ (similar to the RELION experiment). We decoded a representative sample for each cluster. The decoded outputs show that morphology group 0 from our results represent the protein densities present in the thylakoid membrane region (Figure 3 E). We also obtained two morphological groups: single-membrane or double-membrane with some portion of protein densities (decoder outputs are shown in Figure 3E). The remaining groups repeat these morphologies, suggesting that overclustering beyond $K = 8$ may not reveal additional heterogeneity.

Thus, our experiments with subtomograms extracted from the thylakoid membrane region of *Chlamydomonas reinhardtii* tomograms demonstrate the ability of our method to identify macromolecular-scale morphologies, a capability not achievable with existing approaches.

**Reproducibility:** For reproducibility, we included the training and inference code and the necessary instructions to execute them, along with sample datasets and models, in the supplementary materials. We further reported the time and memory requirement of our method and the related methods in

Table 2 in the Appendix. Table 2 shows that our method is superior to other methods in terms of time and memory requirements. Moreover, it also indicates that the integration of MCL module does not result in much additional cost in terms of memory or time.

# 6 DISCUSSION

Identifying the *in situ* morphology of macromolecules from cellular cryo-ET images is extremely difficult given the small size of macromolecules compared to the large cryo-ET images and several other challenges discussed before. The cryo-ET community mostly used manual identification or maximum-likelihood based RELION 3D classification (Scheres, 2012) by manually setting a large number of parameters to identify the macromolecular morphologies. Despite the manual efforts, these approaches would overlook many rare but important macromolecular morphologies. Our work pioneers as a fully-automated, practical solution to identify macromolecular morphologies inside the cell, that is capable of identifying morphologies the other approaches could not.

Being a deep learning based solution, our method also has a few obvious limitations. Though our method is unsupervised and does not require external labels, it is a learning-based solution and requires several thousands of subtomograms to effectively learn the morphologies. For scenarios where only tens or hundreds of subtomograms are available for desired structures, our approach, like any learning-based solution, will not be suitable. Nevertheless, with the help of probabilistic or learned SO(3) priors, this issue can be largely mitigated in the future. In addition, like any unsupervised classification approach (Scheres, 2012; Zivanov et al., 2022; Zeng et al., 2023), our method requires an estimate of the number of classes $K$ to be provided by the user that it uses during the GMM-based latent space clustering step. While applying the method to experimental cryo-ET datasets, providing a high value for $K$ is recommended to ensure that all heterogeneous morphologies are captured. While this may introduce duplicate clusters, such redundancy is acceptable for a comprehensive morphology analysis.

# 7 CONCLUSION

In this paper, we developed a novel unsupervised SE(3) disentanglement method that enables morphology identification of macromolecular complexes from cellular cryo-ET subtomograms. Our method is specifically tailored to solve the inverse problem of mamcromolecular morphology identification with subtomogram-specific method design and a novel multi-choice learning loss. Unlike the existing decade long maximum-likelihood based solution, our method is fully automated and does not miss out rare but crucial morphologies. Our extensive experiments on simulated and experimental cellular cryo-ET subtomogram data validates this claim. We anticipate that our morphology analysis method, being coupled with the downstream subtomogram averaging or subtilt-reconstruction step can determine previously unknown structures with a higher resolution achievable than before. Given the remarkable growth of cellular cryo-ET data collections recently (Last et al., 2025), we foresee our method enabling the study of macromolecular morphology across cell populations, discovering novel biological insights on disease mechanisms and drug response.

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

# A  APPENDIX

## A.1  SINGLE-PARTICLE CRYO-EM VS CELLULAR CRYO-ET

Single-particle cryo-EM collects 2D projection images of purified and isolated macromolecules that are randomly oriented and well separated in a thin layer of vitreous ice. Because many, nearly identical particles are imaged, the resulting micrographs contain relatively high contrast, more uniform backgrounds, and numerous projections of particles with nearly identical morphologies. With 'particle picking' or macromolecule localization, thousands of 2D projection images, each capturing a unique or slightly heterogeneous (different conformations) macromolecular structure with unknown camera angles, are extracted. The projection images are reconstructed into either a single consensus homogeneous structure or a series of structurally heterogeneous conformations. The latter is often referred to as 'heterogeneous reconstruction' in single-particle cryo-EM. The relatively higher SNR and the absence of surrounding cellular material enable near-atomic-resolution reconstructions. CryoDRGN (Levy et al., 2025), CryoSPARC (Punjani et al., 2017), CryoAI (Levy et al., 2025), CryoSPIN (Shekarforoush et al., 2024), etc., all performs homogeneous or heterogeneous reconstruction of 3D structures from 2D single-particle cryo-EM.

On the other hand, cellular cryo-ET collects a tilt series of 2D images through thick, crowded cellular specimens, where each projection contains overlapping densities from membranes, cytoskeleton, organelles, and macromolecular complexes. The 2D tilt-series images are reconstructed into a large 3D grayscale volume, known as a tomogram. The 3D tomograms exhibit extremely low SNR, dramatic contrast attenuation at high tilt angles, and structural clutter that makes macromolecule identification and alignment substantially harder. Unlike single-particle EM micrographs, tomograms also suffer from the missing wedge, an angular region of uncollected data that leads to anisotropic resolution and elongation artifacts (discussed in detail in the following sections). Moreover, while single-particle EM images a homogeneous population of particles that may differ only in conformation, cryo-ET reveals a highly heterogeneous molecular landscape, with both high degrees of compositional and conformational heterogeneity.

The high degrees of compositional heterogeneity present in cryo-ET images make the identification of macromolecular morphology extremely difficult, and often impractical to do directly from 2D projection or tilt-series images. In Figure 4, we provide a fundamental example describing why 2D projection can be misleading to identify 3D object morphology. If a cylinder and a sphere are imaged from the top, both would appear as circles in their corresponding projection images. Thus it is impractical to distinguish between the 3D morphologies just based on the projection image. Hence, cryoDRGN-ET (Powell & Davis, 2024) series of models that reconstruct 3D structures from sub tilt-images are not suitable for identifying compositionally heterogeneous 3D morphologies *in situ*. Instead of projection images, it is practical to classify the 3D images to identify the 3D morphologies. Consequently, the morphologies are identified from 3D subvolumes (often called subtomograms) extracted from the 3D cryo-ET tomograms instead of the 2D tilt-series projection images.

## A.2  CRYO-ET IMAGE ANALYSIS PIPELINE

In cryo-ET, a cellular sample or portion of a cellular sample is imaged with an electron microscope. The sample is tilted up to a certain range at both directions (typically $-60°$ to $+60°$) and an image is captured at each titled position (Turk & Baumeister, 2020). The tilt series images are then back-projected and reconstructed into a 3D voxel image, which is called a tomogram. These tomograms contain *in situ* visualizations of macromolecules and organelles inside a cell and their native spatial organization. However, this unique aspect of tomograms comes with several costs. To maintain the native context of the sample specimen, the electron dosage needs to be kept very low. Due to this low electron dose and also because of the complex cytoplasmic environment, the tomograms become very noisy. Tomograms are also usually very large (e.g., $4000 \times 6000 \times 1000$ voxels) and can not be processed as a whole. Even after binning 4 times across each axis, a tomogram is still large (e.g., $1000 \times 1500 \times 250$ voxels). Each tomogram contains hundreds to thousands of macromolecule, each occupying a miniscule portion of the tomogram. Consequently, the process for macromolecular morphology identification from tomograms occur at the subtomogram level, where a subtomogram is a small subvolume of a tomogram that potentially contains a single macromolecule. Subtomograms are extracted from 3D tomograms using automated particle picking methods (Uddin et al., 2025a;

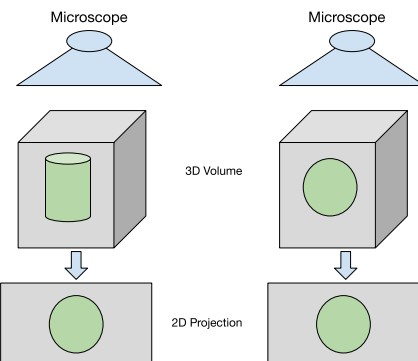

Figure 4: The image shows why projection-image-based reconstruction methods (cryoDRGN (Powell & Davis, 2024) and its variants) are not suitable for highly heterogeneous 3D structure identification.

Liu et al., 2024; Tang et al., 2007) or by manual picking. The extracted subtomograms are then classified and initial coarse 3D templates are generated. RELION 3D classification and our method perform this step. DISCA (Zeng et al., 2023) only classifies the subtomograms, and depends on RELION refinement to obtain the coarse templates. The coarse templates are further refined with subtomogram averaging or subtilt reconstruction to obtain fine-grained and higher-resolution 3D template structures or morphologies. The whole pipeline and the positioning of our method relative to this pipeline are depicted in Figure 5. The figure also contains a schematic diagram visualizing the whole process.

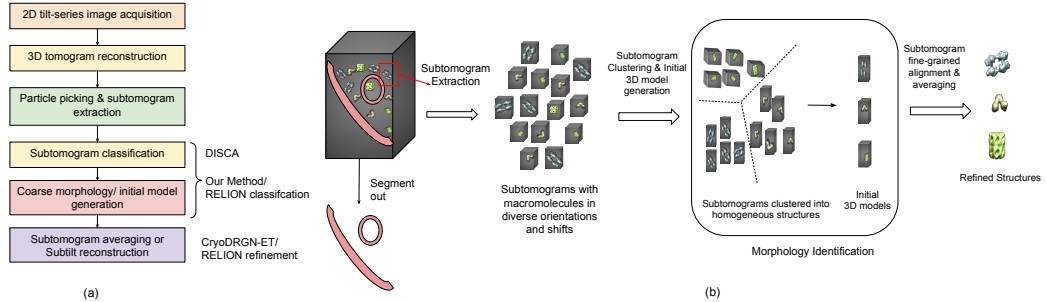

Figure 5: (a) The existing cryo-ET image processing pipeline and our method's positioning, (b)Schematic diagram of identifying refined macromolecular templates from 3D cellular cryo-ET tomogram

### A.3 MISSING WEDGE EFFECT IN CRYO-ET SUBTOMOGRAMS

In cryo-electron tomography (cryo-ET), a cellular specimen is imaged by tilting it incrementally under the electron beam to acquire a series of 2D tilt-series images. However, due to physical and technical limitations of the microscope stage, the tilt range is restricted—typically to about $\pm 60$–$70°$—instead of the full $\pm 90°$. Once the 2D tilt-series images are reconstructed into a 3D tomogram, the incomplete angular coverage during the image acquisition process leaves a wedge-shaped region in the Fourier space of the tomogram unmeasured, known as the missing wedge effect.

Subtomograms extracted from the reconstructed tomogram also carry out the missing wedge effect of the tomograms. Due to the missing wedge effect, subtomograms exhibit anisotropic resolution, with features elongated or distorted along the beam (Z) axis. This elongation affects both structural interpretation and subsequent computational analyses such as alignment, averaging, and classification.

## A.4 EXPERIMENTS

### ENCODER-DECODER NETWORK IMPLEMENTATION

Our encoder comprises four 3D convolutional layers with exponentially linear unit (ELU) activations. The feature maps are progressively downsampled by strided convolutions (kernel size 4, stride 2 for the first three layers; stride 1 for the final layer), followed by two fully connected layers. The output layer produces a concatenated vector containing rotation parameters (three Euler angles), translation offsets, and a latent embedding vector. During training, a dropout layer ($p = 0.2$) is applied to improve generalization. The decoder reconstructs 3D volumes from the latent embedding using a fully connected layer followed by four transposed 3D convolution layers with ELU activations for the first three layers. This sequence progressively upsamples the latent representation back to the original ($48 \times 48 \times 48$) voxel resolution. Dropout ($p = 0.2$) is applied to the fully connected layer during training.

### PREPROCESSING SUBTOMOGRAMS

For preprocessing the subtomograms, we first low-pass-filter the them to 15 Åwith EMAN2. Balancing the trade-off between computational requirement and resolution, we use a box size of 48. To reshape the low-pass-filtered subtomograms to a box size of 48, we performed Fourier space cropping. We used these filtered and reshaped subtomograms, each of size $48 \times 48 \times 48$, to train our model. Before training, we also standardize the intensity of each subtomogram to a mean of 0 and a standard deviation of 1. Upon standardizing the intensities, we applied a soft-edged spherical mask to the subtomograms. The mask is centered within the $48 \times 48 \times 48$ volume with a radius of 24.

### $SO(3)$ GRID SAMPLING FOR MCL:

For the initial epochs ($\approx 40$) of training, we sample the entire $SO(3)$ grid for our MCL loss. After that, we sample near the identity matrix for $SO(3)$. For ease of implementation, we implement this by uniform sampling of 3D axis angles within a fixed range and then converting to $SO(3)$ from the axis angles. For initial epochs (first 40), we uniformly sample axis angles in range $[-90°, 90°]$. We converted 64 axis angle vectors in this range to SO(3) grid and visualized it in Figure 6A. It can be observed that the vectors covered the whole SO(3) grid suggesting our sampling to be correct. After the initial (40) epochs, we uniformly sample axis angles in range $[-30°, 30°]$. We again converted 64 axis angle vectors in this range to SO(3) grid and visualized it in Figure 6B. This time, it only covered region close to the center of the SO(3) grid, that represents the identity matrix. Thus it ensures close to identity SO(3) is sampled after the initial epochs.

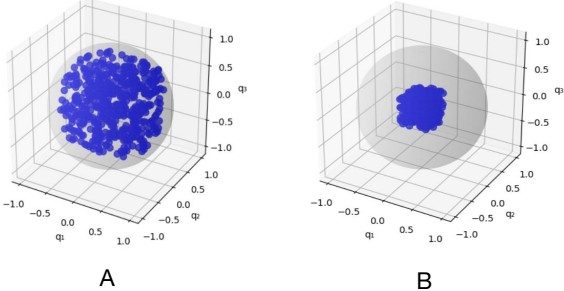

Figure 6: A. Sample $SO(3)$ transformations used for initial epochs of training with MCL loss. B. Sample $SO(3)$ transformations used after initial epochs of training with MCL loss. The samples are plotted on the $SO(3)$ sphere grid.

## A.5 OUR MCL MODULE VS CRYOSPIN MCL MODULE

In Figure 7, we demonstrate the difference between the MCL module in CryoSPIN (Shekarforoush et al., 2024) and the MCL module in our method. In the work by Shekarforoush et al. (2024), the

encoder generates four SO(3) candidates, all of which are used to transform the Fourier decoded volume. The projection images of the transformed Fourier volumes are compared with the input 2D image to compute the MCL loss. The SO(3) candidates are produced by four different encoder heads, and backpropagation is propagated through all the heads. In our work, the candidate SO(3) transformations to transform the decoded volume are not generated by the encoder; rather, they are sampled from the SO(3) grid. We do not backpropagate through the sampling process; instead, we optimize the network with the MCL loss. Furthermore, in terms of method architecture, ours is significantly different from Shekarforoush et al. (2024).

We also experimented by integrating the MCL module of Shekarforoush et al. with the Harmony framework for our realistic simulated dataset. We observe that performance further degrades compared to the original Harmony framework. This is partly due to the additional complexity introduced by using four additional heads to the 3D encoder of the Harmony framework. In addition, the input to our encoder is extremely noisy 3D volumes with missing wedge artifacts; it is difficult to extract the right SO(3) candidates for transforming the decoder volume with heads attached to this encoder and backpropagating through it.

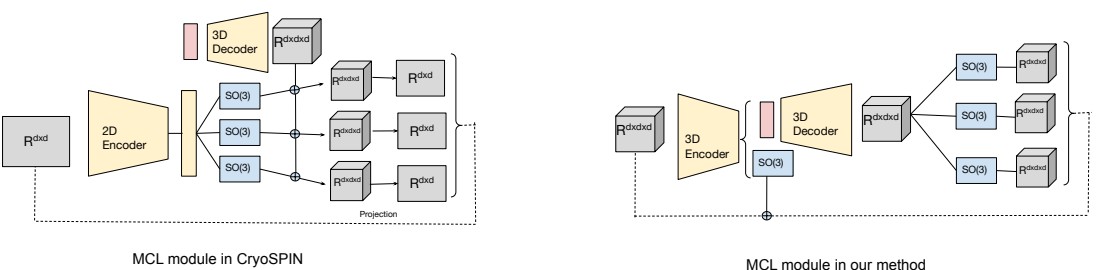

Figure 7: Difference between CryoSPIN MCL module and our MCL module.

EVALUATION METRICS:

**ARI:** To measure the clustering performance, we used Adjusted Rand Index (ARI). The ARI is commonly used to measure the similarity between two data clusterings, correcting for chance. Given a contingency table where:

- $n_{ij}$ is the number of objects in both cluster $i$ of the ground truth and cluster $j$ of the predicted labels,
- $a_i = \sum_j n_{ij}$ is the sum over row $i$,
- $b_j = \sum_i n_{ij}$ is the sum over column $j$,
- $n$ is the total number of data points.

The ARI is defined as:

$$\text{ARI} = \frac{\sum_{ij} \binom{n_{ij}}{2} - \frac{\sum_i \binom{a_i}{2} \sum_j \binom{b_j}{2}}{\binom{n}{2}}}{\frac{1}{2}\left[\sum_i \binom{a_i}{2} + \sum_j \binom{b_j}{2}\right] - \frac{\sum_i \binom{a_i}{2} \sum_j \binom{b_j}{2}}{\binom{n}{2}}}$$

To calculate ARI, we used the ADJUSTED_RAND_SCORE function from SKLEARN.METRICS.

**AUC-FSC:** To evaluate the quality of template morphologies obtained with RELION or our method, we used AUC-FSC (Area Under the Curve of Fourier Shell Correlation), which has been adopted by the community to measure structure recovery performance in heterogeneous cryo-EM/ET datasets (Jeon et al., 2024). It is a scalar metric used to summarize the overall agreement between two 3D volumes in frequency space.

The **Fourier Shell Correlation (FSC)** measures the correlation between two 3D volumes in Fourier space as a function of spatial frequency $s$. It is defined as:

$$\text{FSC}(s) = \frac{\sum\limits_{i \in s} F_1(i) \cdot F_2^*(i)}{\sqrt{\sum\limits_{i \in s} |F_1(i)|^2 \cdot \sum\limits_{i \in s} |F_2(i)|^2}}$$

where:

- $F_1(i)$ and $F_2(i)$ are the complex Fourier coefficients of the two volumes,
- $F_2^*(i)$ is the complex conjugate of $F_2(i)$,
- $i \in s$ denotes the voxels in the shell corresponding to spatial frequency $s$.

The **AUC-FSC** summarizes the FSC curve over the full frequency range $[0, 1]$ and is defined as:

$$\text{AUC-FSC} = \int_0^1 \text{FSC}(s)\, ds$$

**SAP score:** To quantify the disentanglement, several metrics exist, e.g., MIG score, $D_{\text{score}}$, SAP score, etc. (Locatello et al., 2019) demonstrated that these metrics are highly correlated. Following Harmony (Uddin et al., 2022), we primarily used SAP score to measure the SE(3) disentanglement. SAP score is also one of the most acceptable metrics by the community (Kumar et al., 2018). SAP score simply denotes the difference between the top two predictivity scores for ground truth factor by individual latent factors. SAP score for SE(3) disentanglement can be defined as follows:

$$\text{SAP}_{\text{score}} = D(c|z) - D(c|\theta)$$

Here, $c$ is morphology label, $z$ is the morphology latent factor, and $\theta$ is the parameters for SE(3) transformation inferred by the encoder. $D(c|z)$ is the predictivity of morphology labels given the morphology latent factor. $D(c|\theta)$ is the predictivity of morphology labels given the SE(3) transformation factors. The predictivity is calculated using a simple linear model. In our case, we used LinearSVC model to measure the predictivity.

## A.6 RESULTS

### A.6.1 TIME AND MEMORY REQUIREMENTS

In Table 2, we provide the average time per epoch or iteration and the memory requirements to execute our method and the related methods on our benchmark simulated datasets. The ($\downarrow$) indicates, the lower the better.

Table 2: Time and memory requirement of our method and the related methods

| Method | Time (GPU hours) ($\downarrow$) | GPU Memory (GB) ($\downarrow$) |
|---|---|---|
| RELION | 0.15 | 20 |
| CryoDRGN-AI-ET | 8.25 | 41 |
| DISCA | 0.60 | 18 |
| Harmony3D | **0.02** | **6** |
| Our Method | *0.05* | *7* |

### A.6.2 ADDITIONAL TEMPLATE MORPHOLOGIES

The template morphologies generated by RELION, Harmony3D (our method without MCL) and our complete method on realistic subtomogram dataset with SNR 0.01 and missing wedge angle 30° is provided in Figure 2. In this Appendix, we further provide the template morphologies obtained by these methods for the 3 other more idealistic simulated datasets. The obtained template morphologies for SNR 0.1 and missing wedge angle 0° is provided in Figure 8. Similarly, Figure 9 and Figure 10 shows the obtained template morphologies for simulated datasets with SNR 0.1, missing wedge angle 30° and SNR 0.01, missing wedge angle 0° respectively.

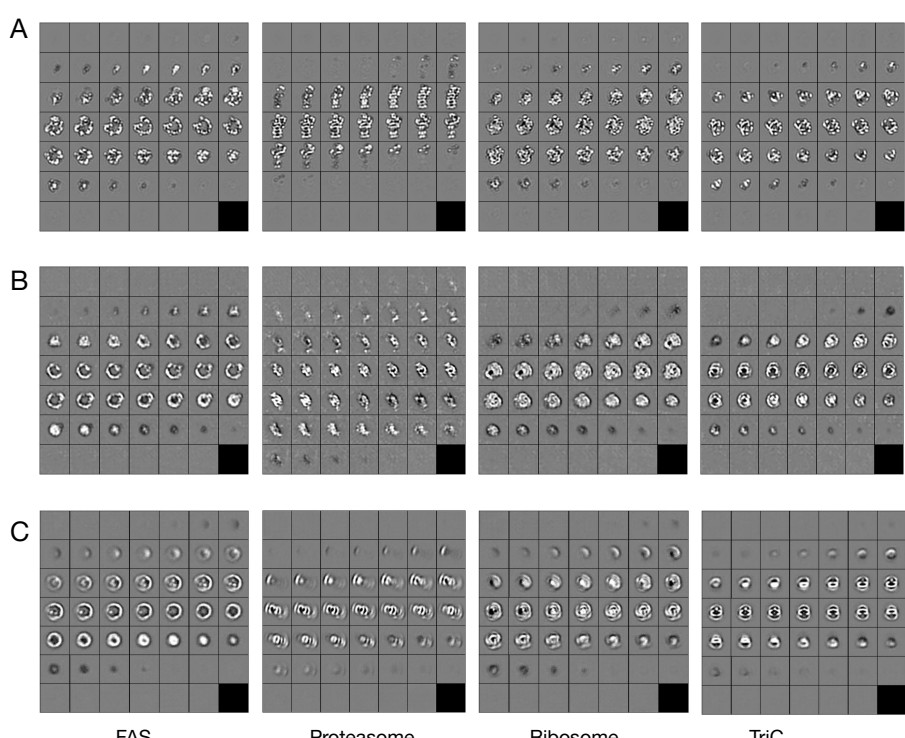

|  FAS | Proteasome | Ribosome | TriC |

Figure 8: The morphology templates obtained by A. RELION, B. Harmony3D, C. Our method on SNR 0.1 missing wedge angle $0°$ simulated dataset

### A.7 STATEMENT ON LARGE LANGUAGE MODEL (LLM) USAGE

Large language models (LLM) were moderately used to improve the clarity and grammar of the manuscript writing. They were not used for any significant tasks, including problem formulation, idea generation, writing from scratch, etc.

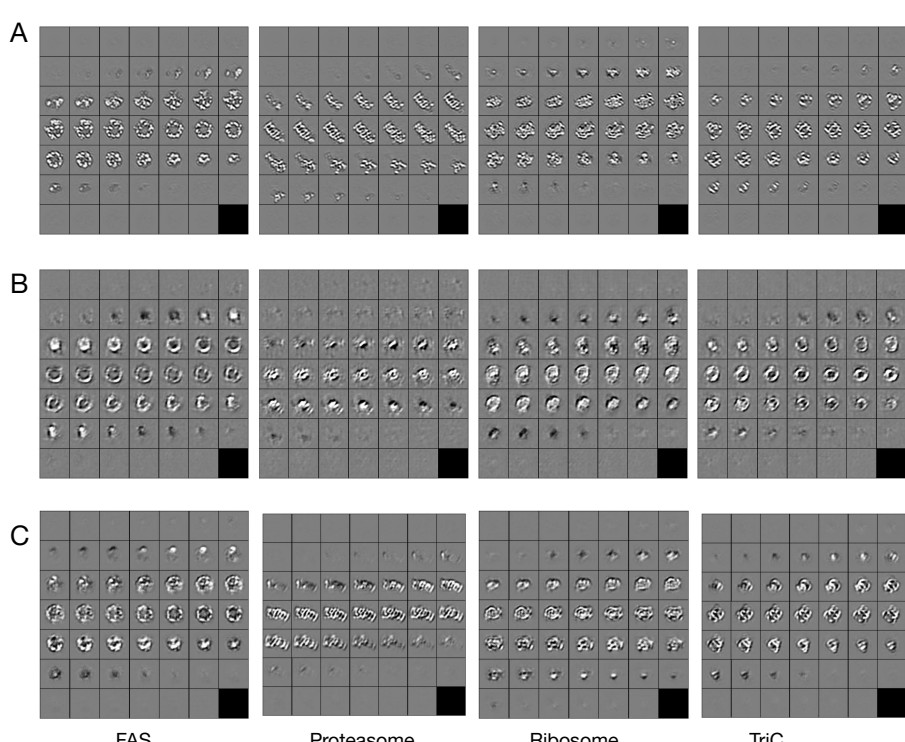

Figure 9: The morphology templates obtained by A. RELION, B. Harmony3D, C. Our method on SNR 0.1 missing wedge angle 30° simulated dataset

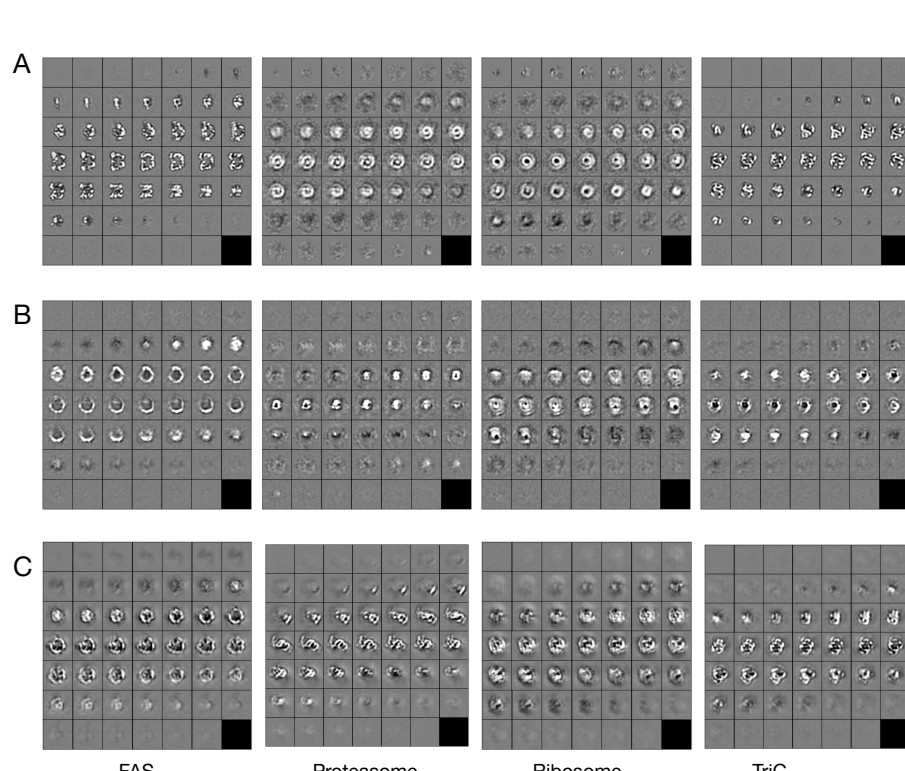

Figure 10: The morphology templates obtained by A. RELION, B. Harmony3D, C. Our method on SNR $0.01$ missing wedge angle $0°$ simulated dataset

