# OpenReview forum: "Unsupervised SE(3) Disentanglement for in situ Macromolecular Morphology Identification from Cryo-Electron Tomography"
_ICLR.cc/2026/Conference — Submitted to ICLR 2026_

### Official Review · Reviewer_trh5 · 2025-10-30

**Soundness:** 2
**Presentation:** 2
**Contribution:** 2
**Rating:** 4
**Confidence:** 3

**Summary:**

This paper proposes an unsupervised deep learning framework for inferring macromolecular morphology from cryo-ET data. It disentangles SE(3) transformations from morphological content and introduces a multi-choice learning module to improve robustness under high noise. The learned morphology representations enable automatic generation of template structures, showing improved performance over traditional methods on simulated and real datasets.

**Strengths:**

1. Learning-based methods for inferring macromolecular morphology from cryo-ET data is an underexplored direction.

2. SE(3) disentanglement, enhanced by multi-choice learning (MCL), provides a reasonable and effective approach to handle transformation and noise in subtomograms.

3. The provision of code and detailed instructions facilitates reproducibility and transparency of the proposed method.

**Weaknesses:**

1. The paper does not provide a fair or sufficiently detailed discussion of how the proposed framework compares to established approaches such as maximum-likelihood or template-matching methods. The narrative focuses mainly on the new framework’s design rather than positioning it clearly within the broader landscape of existing cryo-ET morphology identification techniques.

2. The experimental section reports relatively few results, relying mainly on simulated datasets and a single real cryo-ET dataset.

3.  The authors state that their method “identified several morphologies previously undiscovered with existing approaches.” However, without quantitative validation or biochemical verification, it is scientifically weak to frame these as discoveries. At best, the method identifies previously unclassified or underrepresented density patterns. In structural biology, discovery typically requires further steps such as subtomogram averaging, fitting to known PDB structures, or biochemical confirmation—none of which are presented here.

4. The paper does not compare its performance against CryoDRGN-ET (Rangan et al., Nature Methods, 2024), a highly relevant state-of-the-art deep approach for in situ structure inference from cryo-ET data.

5. Missing references include Levy et al., “Amortized Inference for Heterogeneous Reconstruction in Cryo-EM” (NeurIPS'22), which addresses heterogeneous ab initio reconstruction using amortized inference for cryo-EM.

**Questions:**

1. What is the breakdown of time, memory, and human involvement required to solve a structure using your method? How does this compare to traditional pipelines in terms of computational cost and manual intervention?

2. How does the proposed framework compare with the traditional cryo-ET processing workflow in terms of methodology and practical outcomes? The manuscript could better clarify which specific stages of the conventional pipeline (e.g., classification, alignment, refinement) are replaced or complemented by the proposed learning-based approach, and what limitations still prevent it from achieving comparable reliability to established methods.

3. Can the morphological templates generated by your model be directly used as starting points for downstream subtomogram averaging or high-resolution refinement?

4. Under what specific conditions does the proposed method fail or become unreliable?

---

> ### Author Response · Authors · 2025-12-03
>
> Thanks for your helpful comments. Our responses to your comments and questions are as follows:
>
> **Positioning in cryo-ET image analysis framework**: Please see the discussion on Global Response.
>
> **Additional results and baselines**: Please see the discussion on Global Response.
>
> **Wording issue**: We replaced “undiscovered” with “unclassified” in the revised manuscript.
>
> **Question Answers**
>
> **Q** *What is the breakdown of time, memory, and human involvement required to solve a structure using your method? How does this compare to traditional pipelines in terms of computational cost and manual intervention?*
>
> Please see the discussion on time and memory on Global Response. For human involvement, our method is more automated than RELION 3D classification, since it does not require manually setting several hyperparameters, as is required in RELION (see the RELION command used for 3D classification). It is as automated as the learning-based tools like DISCA, CryoDRGN-AI, etc.
>
> **Q** *How does the proposed framework compare with the traditional cryo-ET processing workflow in terms of methodology and practical outcomes? The manuscript could better clarify which specific stages of the conventional pipeline (e.g., classification, alignment, refinement) are replaced or complemented by the proposed learning-based approach, and what limitations still prevent it from achieving comparable reliability to established methods.*
>
> Please see the discussion on Global Response.
>
> **Q** *Can the morphological templates generated by your model be directly used as starting points for downstream subtomogram averaging or high-resolution refinement?*
>
> Yes. Actually, in our revised manuscript, we used the morphological templates by our model for high-resolution refinement and reported the results in Table 1.
>
> **Q** *Under what specific conditions does the proposed method fail or become unreliable?*
>
> We have clarified this in the discussion section of the manuscript. If any structural class is present in only a few instances compared to the others, or overall, the proposed method may not be able to identify that structure. However, this is a standard limitation of all the relevant techniques.

---

### Official Review · Reviewer_EGtC · 2025-10-31

**Soundness:** 2
**Presentation:** 3
**Contribution:** 3
**Rating:** 6
**Confidence:** 3

**Summary:**

This paper tackles the challenge of identifying macromolecules in cryo-ET. The most traditional approach transforms macromolecule identification into estimating SE(3) and a template, using statistical methods such as maximum likelihood estimation. However, this inevitably leads to insensitivity to rare features. This paper continues the line of thought on problem transformation, proposing a representation learning framework instead of statistical methods to estimate the template. Key techniques in this method include unsupervised SE(3) disentanglement and multi-choice learning. Experiments demonstrate that the proposed method significantly improves performance compared to traditional methods, especially in low signal-to-noise ratio scenarios. Finally, the paper uses the proposed method to reveal previously unidentified macromolecular morphologies.

**Strengths:**

1. The paper constructs multiple simulated subtomogram datasets of macromolecular mixtures that should be valuable for subsequent research.
2. The work reveals previously unidentified macromolecular morphologies, underscoring its practical significance.
3. Compared with traditional methods, the proposed approach demonstrates clear performance gains and addresses a decade-long unsolved problem in structural biology.

**Weaknesses:**

1. Limited novelty.
The method comprises two key components: unsupervised SE(3) disentanglement and multiple-choice learning. The reconstruction and regularization loss used for unsupervised SE(3) disentanglement are common loss. Moreover, Shekarforoush et al. have previously applied the MCL idea to estimate SO(3), so I suppose using MCL here to estimate SE(3) templates does not show enough novelty.

2. Missing baselines.
Beyond the classical maximum-likelihood method RELION, this paper does not compare against any other baselines  (Harmony3D is essentially an ablation targeting MCL rather than an independent baseline). Since the Introduction notes that this problem has been investigated for more than a decade, the limited benchmarking makes the claimed performance improvements less convincing.

3. Potentially unfair comparison.
MCL significantly increases the computational cost during the forward pass. However, the quantitative comparison in Table 1 does not report training time or memory usage for the baselines, making it difficult to assess the fairness of the results.

**Questions:**

1. Please clarify further the differences between your method and that of Shekarforoush et al.
2. Please report the baseline’s training time and memory consumption as used in the paper.
3. Please clarify the choice of baselines and, if possible, evaluate more.

---

> ### Author Response · Authors · 2025-12-03
>
> Thank you for your helpful comments. Please find our responses to your comments below:
>
> **Novelty**: Please see the discussion on Global Response
>
> **Baselines**: Please see the discussion on Global Response
>
> **Training time & memory**: Please see the discussion on Global Response
>
> **Question Answers**:
>
> **Q** *Please clarify further the differences between your method and that of Shekarforoush et al.*
>
> Clarified in the Global Response.
>
> **Q** *Please report the baseline’s training time and memory consumption as used in the paper.*
>
> Please see the discussion on Global Response.
>
> **Q** *Please clarify the choice of baselines and, if possible, evaluate more.*
>
> Clarified in the Global Response. More baselines were evaluated.

---

### Official Review · Reviewer_Dg9Q · 2025-10-31

**Soundness:** 2
**Presentation:** 2
**Contribution:** 1
**Rating:** 2
**Confidence:** 4

**Summary:**

This paper proposes an unsupervised deep-learning framework for identifying 3D macromolecular morphologies directly from noisy cryo-electron-tomography (cryo-ET) subtomograms. The method disentangles SE(3) rigid-body transformations from morphological content using a modified Harmony framework (Uddin et al., CVPR 2022), combined with a new Multi-Choice Learning (MCL) module. The MCL component introduces multiple transformed decoder outputs and applies a winner-takes-all loss to enhance robustness under extremely low SNR conditions. Experiments on simulated mixtures of macromolecules (ribosome, proteasome, TRiC, FAS) and real Chlamydomonas reinhardtii tomograms demonstrate improved clustering (ARI ≈ 0.85–1.0) and better template fidelity (AUC-FSC ≈ 0.25 vs ≈ 0.15 for RELION). Qualitative results (Fig. 3) suggest discovery of new thylakoid-membrane morphologies not captured by previous approaches.

**Strengths:**

1. Important problem domain: fully automated, unsupervised morphology identification from in-cell tomograms.

2. Empirical robustness: large ARI/AUC-FSC gains under extreme noise conditions.

3. Elegant MCL integration: computationally light mechanism improving convergence.

4. Comprehensive evaluation: simulated + real datasets; thorough appendix for reproducibility.

**Weaknesses:**

1. Limited novelty – no new theoretical insight beyond adapting Harmony + MCL. The proposed model inherits almost the entire architecture and loss design from Harmony (CVPR 2022) and Multi-Choice Learning (Guzman-Rivera 2012; Kohl 2018). The paper provides no new theoretical analysis of SE(3) disentanglement or identifiable latent factors. Equations (4)–(7) merely restate known forms with SE(3) transforms substituted for generic transformations. There is no formal proof that the new objective achieves rotation- or translation-invariant representations, nor any mathematical link between MCL and reduced latent-space entanglement. Consequently, the method reads more as an engineering adaptation of Harmony to volumetric cryo-ET data rather than a conceptual advance in disentanglement theory.

2. Missing modern baselines – no comparison to CryoDRGN-ET (2024) [1], CryoSPIN (2024) [2], or DISCO-ET (2023) [3]. The paper evaluates only against RELION and Harmony3D, both of which predate recent breakthroughs in deep cryo-ET analysis. Newer baselines such as CryoDRGN-ET ( Rangan et al., Nature Methods 2024 ), CryoSPIN ( Shekarforoush et al., NeurIPS 2024 ), and DISCO-ET ( Zeng et al., PNAS 2023 ) provide neural or hybrid Bayesian formulations capable of capturing continuous conformations and pose distributions—precisely the challenges this paper claims to address. Without head-to-head quantitative comparisons, it is impossible to assess whether the proposed approach actually advances the state of the art or merely improves upon legacy maximum-likelihood methods. Given that CryoDRGN-ET and CryoSPIN already demonstrate SE(3)-aware latent modeling with significantly higher AUC-FSC and visual quality, the omission seriously undermines the empirical credibility of the claims.

3. Weak quantitative results on real data – evaluation remains descriptive. Although Table 1 shows substantial gains on synthetic benchmarks, the real-data evaluation (Fig. 3) is largely qualitative. The reported AUC-FSC values (< 0.3) indicate coarse reconstructions that fall short of practical interpretability in structural biology. There are no statistical tests, confidence intervals, or multiple-run variances to demonstrate robustness. Moreover, the claimed “discovery” of novel morphologies is supported only by visual inspection without independent biological validation or cross-dataset reproducibility. In high-impact imaging applications, descriptive results are insufficient; quantitative verification using expert annotations or known templates is necessary.

4. High data requirement – dependence on thousands of subtomograms limits practicality. The method’s training pipeline relies on tens of thousands of subtomograms to stably estimate SE(3) latent factors. This dependence severely limits usability in many biological contexts, where obtaining even hundreds of particles is challenging due to experimental constraints. The authors mention no strategies for data-efficiency, pretraining, or transfer learning. In contrast, modern probabilistic approaches such as CryoDRGN-ET can operate effectively with one or two orders of magnitude fewer examples by leveraging pose amortization. Thus, while the model performs well in controlled simulations, it remains unrealistic for many real-world cryo-ET studies.

5. No latent-space disentanglement metrics – rotation/translation invariance not numerically verified.  The paper’s central claim is SE(3) disentanglement, yet it provides no direct quantitative evidence. There is no metric assessing how invariant latent representations remain under known rotations or translations. Standard evaluations—e.g., mutual-information reduction between pose and content variables, linear-probe predictability of rotation angles, or equivariance correlation tests—are entirely absent. Without such analysis, it is unclear whether the latent space truly isolates morphology from pose, or if improved clustering simply arises from data augmentation. This omission weakens the paper’s main theoretical claim and leaves the effectiveness of the proposed representation ambiguous.

[1] CryoDRGN-ET: Deep reconstructing generative networks for visualizing dynamic biomolecules inside cells. Nature Methods. 2024.

[2] CryoSPIN: Improving ab initio cryo-EM reconstruction with semi-amortized pose inference. NeurIPS. 2024.

[3] DISCO-ET. High-throughput cryo-ET structural pattern mining by unsupervised deep iterative subtomogram clustering. 2023.

**Questions:**

1. Can you report quantitative SE(3) invariance metrics for the latent space?

2. Why exclude CryoDRGN-ET, CryoSPIN, and DISCO-ET from comparisons?

3. How sensitive is performance to the number of transformation samples N in MCL formulation?

4. Could probabilistic or learned SO(3) priors reduce the large-data requirement?

5. How does MCL differ mathematically from standard mixture-of-experts formulations?

---

> ### Author Response · Authors · 2025-12-03
>
> Thank you for your insightful comments regarding our paper. Please find our responses to your comments below:
>
> **Novelty**: Please see the discussion on Global Response
>
> **Missing modern baselines**: Please see the clarification on Global Response
>
> **Weak quantitative results**: Our classification performance (ARI, Acc) is very strong and sufficient for classifying morphologies within subtomograms. For AUC-FSC, we previously reported only the values for “coarse templates” obtained by our method; hence, the AUC-FSC looks lower. These “coarse templates” can be refined further to improve the AUC-FSC. To demonstrate this, we also refined the “coarse templates” with the subtomogram averaging or template refinement step in RELION and reported the final AUC-FSC values. All of these values are sufficiently high to identify the structures.
>
> **High data requirement**: We would respectfully disagree with the remarks. In our simulated dataset, we have 1000 subtomograms per morphology class. This is very low compared to what other relevant learning based methods are used. CryoDRGN-ET and CryoDRGN-AI were tested against sub-tilt images of 119,000 subtomograms for one morphology class to resolve different conformations, which is several magnitudes higher than ours.
>
> **Latent space disentanglement metrics**: Please see the discussion on Global Response.
>
> **Question Answers**:
>
> **Q** *Can you report quantitative SE(3) invariance metrics for the latent space?*
>
> Answered in Global Response.
>
> **Q** *Why exclude CryoDRGN-ET, CryoSPIN, and DISCO-ET from comparisons?*
>
> Answered in Global Response.
>
> **Q** *How sensitive is performance to the number of transformation samples N in the MCL formulation?*
>
> Thanks for the insightful question. N determines the number of candidates in the MCL module. In our experiments, we used N=96. If we used N=1, it would have reduced to the Harmony3D model, since there is only one candidate. Overall, we found that $N\geq64$ gave reasonable performance under the SNR 0.01 setting. For higher SNR idealistic datasets, much lower N is sufficient.
>
> **Q** *Could probabilistic or learned SO(3) priors reduce the large-data requirement?*
>
> First, we do not have a larger than usual data requirement, as we clarified previously. Having probabilistic or learned SO(3) priors may further reduce the amount of data required or the value of N needed to achieve a feasible level of performance. We have added it to the Discussion section of the revised manuscript.
>
> **Q** *How does MCL differ mathematically from standard mixture-of-experts formulations?*
>
> Standard mixture-of-experts minimizes a weighted average of multiple expert losses:
> $$\sum_{i=1}^M w_i\, \ell_i.$$
>
> MCL minimizes the best expert's loss:
> $$\min_{i \in \{1,\dots,M\}} \ell_i.$$
>
> Here, $M$ is the total number of experts in the model. $w_i$ is the weight for expert $i$, produced by a gating network, with
> $w_i \ge 0,  \sum_{i=1}^M w_i = 1$.
>
> $\ell_i$ is the loss incurred by expert $i$ on a given training example $(x, y)$: $\ell_i = \ell\big(y, f_i(x)\big)$ and $f_i$ is the $i$-th expert.

---

### Official Review · Reviewer_AWRh · 2025-11-01

**Soundness:** 2
**Presentation:** 1
**Contribution:** 2
**Rating:** 2
**Confidence:** 5

**Summary:**

The paper introduced a method to identify (or classify or cluster, more accurately) subtomograms without pose alignment by unsupervised SE(3) disentanglement. Multi-choice learning is used in this method to greatly improved the performance in low SNR samples. The authors showed the performance of the proposed method and other baselines on both synthetic and real experimental data.

**Strengths:**

- The classification of subtomograms is a very important, yet difficult problem. Instead of optimizing subtomogram poses and classes at the same time (as in many traditional methods), the paper tried to directly disentangle SE(3) transformation and classify/cluster in the latent space.

- The introduction of the MCL module is interesting and greatly improved the performance.

**Weaknesses:**

- The presentation of the results can be much improved. The slice-by-slice visualization is hard to see or interpret and I would suggest the authors to only use the isosurface visualization of the 3D density maps. With the current presentations, it is extremely difficult to evaluate the performance of the propose method. I would increase my score if the presentation is improved and demonstrates a reasonable performance of the proposed method.

- The paper also lacks novelty from the method perspective, as most of the methodology is build on the Harmony framework, except the MCL module.

**Questions:**

- How was the "template" of Relion's 3D classification derived? Are they the result of the subtomogram averaging after classification? Did Relion's 3D classification also optimize the poses of the subtomograms? If yes, this is not a fair comparison since the proposed method in the paper did not optimize the poses.

- Do all subtomograms share the same CTF?

---

> ### Author Response · Authors · 2025-12-03
>
> Thank you for your valuable comments regarding our paper. We provide responses to your comments in the following:
>
>
> **Presentation**: We have improved the presentation of the figures (Figure 2 and Figure 3) by using only isosurface visualization.
>
> **Novelty**: Please see Global Response.
>
>
> **Question Answers**:
>
> **Q** *How was the "template" of Relion's 3D classification derived? Are they the result of the subtomogram averaging after classification? Did Relion's 3D classification also optimize the poses of the subtomograms? If yes, this is not a fair comparison since the proposed method in the paper did not optimize the poses.*
>
> The “template” of Relion’s 3D classification is the resultant initial 3D volumes that RELION uses for further refinement (subtomogram averaging). We use the following command to perform RELION 3D classification:
>
> `which relion_refine --o Class3D/job001/run --i all_images.star --ref prior.mrc --firstiter_cc --dont_combine_weights_via_disc --pool 4 --pad 2 --iter 50 --tau2_fudge 4 --particle_diameter 360.00 --K 4 --flatten_solvent --zero_mask --oversampling 1 --healpix_order 2 --offset_range 6 --offset_step 2 --norm --scale --j 48 --ref_angpix 7.50 --gpu 2,3,4 --grad`
>
> Upon execution of the command, RELION classifies the images in the dataset (mentioned in the .star file) into K clusters and also generates a “template” structure for each cluster. These templates are used for fine-grained refinement in the latest subtomogram averaging step.
>
> We would like to clarify a misunderstanding. The encoder network of our method outputs poses per subtomogram relative to the decoded template. Our SE(3) disentanglement framework also optimizes the poses. Thus, our method also does coarse refinement as Relion’s 3D classification, hence they are a fair comparison.
>
> **Q** *Do all subtomograms share the same CTF?*
>
> Yes, in the experiments, all subtomograms share the same CTF.

---

### Author Response · Authors · 2025-12-03
**Global Response**

We would like to thank the reviewers and the area chairs for taking the time to review our paper. We are glad that the reviewers found our work to be important (AWRh, Dg9Q, EGtC, trh5), empirically robust (Dg9Q, EGtC), and our integration of the MCL module in disentanglement to be elegant (AWRh, Dg9Q, trh5).

Here, we address several of the common concerns shared by the reviewers. We address individual concerns in individual responses.

**Methodological Novelty**: Our MCL module significantly improved the optimization of the 3D convolutional neural network for SE(3) disentanglement. This is a novel and elegant addition to the Harmony disentanglement framework (**Dg9Q**) and greatly improves the task performance (**AWRh**). Furthermore, our MCL module is novel and significantly different from the MCL module used in single-particle cryo-EM reconstruction work by CryoSPIN or Shekarforoush et al. (as clearly demonstrated in Figure 7 in the Appendix). In the work by Shekarforoush et al., the encoder generates four SO(3) candidates, all of which are used to transform the Fourier decoded volume. The projection images of the transformed Fourier volumes are compared with the input 2D image to compute the MCL loss. The SO(3) candidates are produced by four different encoder heads, and backpropagation is propagated through all the heads. In our work, the candidate SO(3) transformations to transform the decoded volume are not generated by the encoder; rather, they are sampled from the SO(3) grid. We do not backpropagate through the sampling process; instead, we optimize the network with the MCL loss. Furthermore, in terms of method architecture, ours is significantly different from Shekarforoushet al. In the Appendix of the revised manuscript, we also discussed why the MCL module in Shekarforoush et al is not suitable in our case.

**Fit into existing cryo-ET processing pipeline**: Our method performs 3D classification and the initial 3D model generation step in the cellular cryo-ET image analysis pipeline. This is similar to RELION’s 3D classification process, which we consider a baseline for fair comparison. We further clarified this with a visualization in Figure 5 of the Appendix.

**Comparison with additional baselines**: CryoSPIN (Shekarforoushet al.) is a method for single-particle 2D cryo-EM image analysis, not 3D cellular cryo-ET image analysis. CryoSPIN, CryoAI, and CryoDRGN all perform heterogeneous reconstruction of 3D volumes or structures from single-particle 2D cryo-EM images, which are much less noisy compared to our 3D cellular cryo-ET images. In the supplementary document of the revised submission, we added a section describing the differences between single-particle cryo-EM and cellular cryo-ET.

CryoDRGN-ET is a method tailored for cryo-ET that performs sub-tilt series reconstruction to obtain structures and is somewhat equivalent to the subtomogram averaging step in the cryo-ET image processing pipeline. In addition, cryoDRGN-ET requires the SO(3) poses to be given as input, which our method or RELION do not. Hence, cryoDRGN-ET would not be a fair comparison. A somewhat fair comparison would be CryoDRGN-AI-ET, which does not require SO(3) poses to be given as input, but performs sub-tilt reconstruction similar to CryoDRGN. To this end, we evaluated CryoDRGN-AI-ET on our datasets and reported the results in Table 1. As expected, CryoDRGN-AI-ET does not perform well in the high-heterogeneity condition of our problem setup. CryoDRGN frameworks are designed for low-heterogeneity conditions, which can reconstruct conformationally diverse structures of the same composition from sub-tilt or projection images. In Figure 4 of the Appendix, we demonstrate why projection images are often not optimal for resolving higher degrees of 3D heterogeneity.

DISCA (which reviewer Dg9Q mentioned as DISCO-ET) is a method for 3D classification in the cryo-ET image analysis pipeline. For initial template generation, DISCA uses RELION’s 3D classification process. We evaluated DISCA against our benchmark datasets and reported the results in Table 1. As shown, DISCA performance also degrades significantly at lower SNR, and it is unable to distinguish several distinct structures in the realistic dataset.

**Disentanglement evaluation**: We quantitatively evaluated the latent space disentanglement performance of our method and Harmony3D using the SAP score and reported the results in Table 1. The scores suggest substantial improvements in our method over Harmony3D, particularly on lower-SNR realistic datasets, primarily due to the integration of our novel MCL module into the training process.


**Time and memory**: We reported the time and memory consumptions of our method and the baseline (both previous and newly added) in Table 2 of the Appendix of the revised manuscript.

---

### Meta-Review · Area_Chair_w7Hd · 2026-01-08

**Summary:**

This work reads as a solid engineering adaptation of an existing disentanglement framework (Harmony) with an added winner-takes-all multi-choice loss, backed by strong synthetic results, but it still does not convincingly clear the “ICLR contribution bar” given (i) persistent novelty concerns, (ii) incomplete head-to-head evaluation vs the most relevant modern deep cryo-ET/pose-inference baselines, and (iii) real-data validation that remains comparatively weak/qualitative relative to the strength of the claims.

Across reviews, the dominant blockers were: limited methodological novelty beyond combining Harmony-style SE(3) disentanglement with MCL; missing/insufficiently fair comparisons to state-of-the-art cryo(-ET) deep methods (especially CryoDRGN-ET / CryoSPIN / DISCO-ET-style lines); and real-data evidence that is not rigorous enough for “discovery” or strong practical claims (need clearer quantitative validation, robustness, and/or external confirmation). Secondary concerns were presentation/figure interpretability and fairness/transparency of compute and pipeline positioning. (These align with the score pattern: two confident rejects, one weak accept, one weak reject.)

**Reviewer Concerns:**

The rebuttal partly addressed several practical concerns by (i) improving figure presentation (isosurface-only) to respond to AWRh’s readability criticism, (ii) clarifying pipeline fit and RELION fairness by arguing their encoder also outputs/optimizes poses (AWRh/trh5), (iii) adding additional baselines (DISCA and CryoDRGN-AI-ET) plus reporting time/memory (EGtC/trh5), (iv) clarifying how their MCL differs from Shekarforoush/CryoSPIN and providing a simple MoE-vs-MCL objective contrast (Dg9Q/EGtC), (v) adding at least one disentanglement-oriented metric (SAP) and refining “undiscovered”→“unclassified” wording (Dg9Q/trh5). However, major issues remain outstanding: the core novelty concern (still largely Harmony + MCL without a new conceptual/theoretical contribution), the absence of direct, convincing head-to-head comparisons against the most relevant modern SOTA (especially CryoDRGN-ET and CryoSPIN/DISCO-ET lines, which are largely argued away rather than empirically settled), and the strength of real-data validation (still limited/qualitative relative to the paper’s biological claims and lacking robust uncertainty/variance or independent validation), along with the request for direct SE(3) invariance/equivariance tests beyond SAP.

**Reviewer Scores:**

Reviewer AWRh (2 → ~4): With the isosurface-only figure revisions and the clarification that the method also estimates poses (making the RELION comparison less obviously unfair), they’d likely raise to a borderline “weak reject / near-threshold,” but still dock novelty (“mostly Harmony + MCL”).

Reviewer Dg9Q (2 → ~4): The added baselines (DISCA, CryoDRGN-AI-ET), compute table, and SAP score would probably move them slightly, but without a true head-to-head vs CryoDRGN-ET/CryoSPIN/DISCO-ET and stronger real-data quantification, they’d remain rejecting.

Reviewer EGtC (6 → ~6): Their main asks (baseline breadth, time/memory fairness, clearer distinction from Shekarforoush) are substantially addressed, so they’d likely strengthen to a clearer accept, though still noting limited novelty.

Reviewer trh5 (4 → ~4): Softening “undiscovered” to “unclassified,” clearer pipeline positioning, and added baselines/compute reporting would likely shift them from weak reject to borderline, but they’d still hesitate due to missing CryoDRGN-ET comparison and limited quantitative real-data validation.

---

### Decision · Program_Chairs · 2026-01-26

Reject